# The Implementation of a Gesture Recognition System with a Millimeter Wave and Thermal Imager

**DOI:** 10.3390/s24020581

**Published:** 2024-01-17

**Authors:** Yi-Lin Cheng, Wen-Hsiang Yeh, Yu-Ping Liao

**Affiliations:** Department of Electrical Engineering, Chung Yuan Christian University, Taoyuan 320, Taiwan; s10525150@cycu.org.tw (Y.-L.C.); peter880127@gmail.com (W.-H.Y.)

**Keywords:** deep neural network, object recognition, millimeter wave radar, thermal imager, gesture recognition

## Abstract

During the COVID-19 pandemic, the number of cases continued to rise. As a result, there was a growing demand for alternative control methods to traditional buttons or touch screens. However, most current gesture recognition technologies rely on machine vision methods. However, this method can lead to suboptimal recognition results, especially in situations where the camera is operating in low-light conditions or encounters complex backgrounds. This study introduces an innovative gesture recognition system for large movements that uses a combination of millimeter wave radar and a thermal imager, where the multi-color conversion algorithm is used to improve palm recognition on the thermal imager together with deep learning approaches to improve its accuracy. While the user performs gestures, the mmWave radar captures point cloud information, which is then analyzed through neural network model inference. It also integrates thermal imaging and palm recognition to effectively track and monitor hand movements on the screen. The results suggest that this combined method significantly improves accuracy, reaching a rate of over 80%.

## 1. Introduction

As the COVID-19 pandemic continued to spread, the virus posed a risk of transmission through various routes, including droplets and direct contact [1]. While the use of alcohol-based disinfectants and hand washing with soap can help reduce the risk of exposure, these methods may not provide complete bacterial isolation. With the rapid growth of the Internet of Things (IoT), 5G communication networks, and self-driving cars, car manufacturers and governments are focusing on creating advanced smart vehicle systems that use remote communication and information technologies [2]. In ref. [3], they introduced a technology that used millimeter wave radar sensors for gesture recognition to control in-vehicle infotainment systems. This technology provides a safe and intuitive control interface that reduces the possibility of driver distraction. This is unlike traditional car control panel buttons that require drivers to look away from the road for short periods. Furthermore, due to global technological advancements and medical breakthroughs, people are living longer than before. As a result, elderly care and assistance have become more important. Gesture recognition applications for elderly care are also introduced in the literature [4]. Different gestures are analyzed by Kinect sensors; each gesture is associated with a specific request, such as water, food, toilets, help, and medicine. It is then sent as a text message to the caregiver’s smartphone to reduce the caregiver’s workload. As a result, there is a growing demand for non-contact control measures. Simultaneously, the rapid advancement of artificial intelligence (AI) was transforming various aspects of our daily lives. Many applications now integrate AI to improve convenience; one such example is the computer vision-based intelligent elevator information system proposed in [5]. This technology uses face recognition to predict passenger characteristics and traffic flow for energy-efficient elevator management. The system uses object recognition technology, such as the YOLO (You Only Look Once) algorithm [6], to identify the user’s face. YOLOv7 is an object detection model that was launched in July 2022. It offers a 50% reduction in computation and a 40% reduction in parameters, which has effectively improved its accuracy and speed. YOLOv7 outperforms most object detectors in terms of accuracy and speed in the range of 5 FPS to 160 FPS [7]. In recent years, gesture recognition has gained prominence in various applications, including virtual reality (VR), human–computer interaction, and sports medicine [8,9,10,11,12]. Traditional gesture recognition is typically based on two primary methods. The first method involves the use of data gloves fitted with sensors capable of detecting finger movements, which then transmit electronic signals to a computer for gesture recognition [13]. However, this approach requires the use of specific hardware, and the sharing of gloves can introduce the risk of virus transmission.

The second method is Vision sensing. Vision sensing has become the prevailing approach to machine vision in recent years, utilizing captured images for analysis. This method transcends the restraints of two-dimensional image capture with the growing implementation of three-dimensional imaging systems utilizing dual-lens or depth cameras in diverse applications [14]. Additionally, 2D images can have issues with background complexity, occlusion, illumination, and fast motion, which can be tackled by 3D cameras with depth information [15]. Contemporary technology for recognizing gestures mainly depends on a lens combined with deep learning. As proposed in [16], one method proposed is the use of optical cameras for capturing and analyzing both dynamic and static gestures. However, this technique requires precise lighting conditions and fails to provide in-depth information. On the other hand, an alternative strategy utilizes RGB-D depth cameras for gesture classification [17]. Despite this, such devices are unsuited for application in direct sunlight. Privacy concerns could arise with optical camera-based gesture recognition systems, since users may be concerned about unauthorized image capture or malicious use without their consent.

By contrast, miniature radar sensors offer a potential solution to overcome the limitations associated with cameras. In ref. [18], they proposed the importance of dynamic gesture recognition, which is an important part of human–computer interaction. The practical application of the gesture recognition system involves recognizing various dynamic and continuous gestures [19]. However, the extraction of gesture features may be affected by changes in ambient light and background. In ref. [20], they proposed using millimeter wave (mmWave) radar for recognizing large-motion gestures. Millimeter wave radar is a radar system that operates in the millimeter wave frequency band, mainly using short-wavelength electromagnetic waves [21]. A linear FM signal, also known as Frequency Modulated Continuous Wave (FMCW), is a type of sine wave signal that has a linearly increasing frequency over time. FMCW millimeter wave radar technology is known for its ability to provide accurate depth information and is less susceptible to temperature changes. It is particularly useful for measuring challenging environments such as occluded areas, foggy conditions, and both indoor and outdoor scenarios [22]. In ref. [23], they also proved the interference immunity of millimeter wave radar by actually performing millimeter wave radar imaging tests in a low-visibility, smoky environment. The use of millimeter wave radar allays privacy concerns. The research in [24,25] laid the foundation for using millimeter waves to recognize gestures. To classify different gestures, deep neural networks have been widely used for multi-class classification tasks [26,27,28]. Google Soli has utilized the range–Doppler (RD) spectrum for gesture recognition using 60 GHz frequency-modulated continuous wave (FMCW) radar sensors. Soli has brought this technology into the context of micro-gesture sensing, wearables, and smart devices. Soli has proposed a 60 GHz millimeter wave FMCW radar for detecting fine-grained gestures, capable of detecting four gestures from a single user. However, the proposed algorithm requires a computing power and is mainly used on PCs. In 2022, we released a millimeter wave-based gesture-controlled smart speaker to solve the privacy problem in smart homes. The smart speaker uses millimeter wave radar and can be redirected to any nearby location with a clapping sound. We can instantly analyze and classify five dynamic arm gestures by running a deep neural network (DNN) on a small but powerful computer, the NVIDIA Jetson Nano development kit. The result of the gesture recognition triggers the corresponding music control [29]. As a result, more research is being conducted on hand gesture recognition using cost-effective and miniature radar sensors. With the advancement of smart camera technology, it has found a wide range of applications, such as surveillance systems, facial recognition, and more [30]. However, there is an increasing concern that cameras may violate people’s privacy by capturing unwanted images. In this regard, Kim proposed a method to protect privacy by blurring the unwanted areas of the image, such as faces [31]. However, the use of cameras raises privacy concerns and may have an impact on the recognition rate in situations where lighting is too bright or insufficient. Amidst the COVID-19 pandemic, numerous organizations have opted to incorporate infrared thermal imaging cameras equipped with advanced AI face detection software to gauge body temperature. These cameras are capable of detecting and measuring infrared radiation energy released from an object’s exterior, altering infrared radiation and the infrared radiation energy distribution into a visual image. Correspondingly, the device facilitates temperature measurements in dimly lit settings without being affected by light-related inconsistencies. Moreover, the data are processed through image processing to produce images exhibiting specific color distributions. This procedure mitigates the risk of data leakage and resolves issues related to optical cameras. This study introduces a gesture recognition system for large motions with a fusion of millimeter wave radar and thermal imagers, along with the integration of deep learning. Millimeter wave radar sensor is typically good at detecting radial motion, while a thermal imager can detect lateral motion; this complementarity promotes the fusion of measurements from both sensors to improve the accuracy of gesture recognition. This ground-breaking approach not only eradicates the necessity for physical contact with devices but also alleviates privacy anxieties linked to facial recognition cameras.

The rest of the paper is organized as follows: In Section 2, we introduce hand image recognition using a thermal imager and point cloud data collection by millimeter wave radar. Additionally, we discuss the processing flow and the neural network training for hand gesture recognition. In Section 3, we present our experimental results and compare the effectiveness of two methods: using millimeter wave alone and using millimeter wave combined with thermal imagers. Section 4 discusses the limitations of this study. Finally, Section 5 provides a summary of the paper.

## 2. Materials and Methods

In our study, we used a Lepton 3.5 thermal imager with a resolution of 160 × 120 pixels to capture hand images [32]. The module is capable of Long-Wave Infrared (LWIR) detection, which detects the infrared radiation energy emitted by an object, converts the energy into temperature, and then creates and displays a color image. Joybien Batman’s BM201-PC3 mm wave radar module [33] is used to collect hand-point cloud information. The millimeter wave module uses a Texas Instruments (TI) IWR6843 mm-wave sensor as its core and FMCW (Frequency Modulated Continuous Wave) radar technology. It operates primarily in the 60 GHz to 64 GHz band and has a continuous FM bandwidth of 4 GHz. The module uses four receive antennas and three transmit antennas, which can be used for range and speed measurement. After collecting the hand data, we use Jetson Xavier NX [34] for data processing and gesture recognition. Jetson Xavier NX is a Small On Module (SOM) system that is only 70 mm × 45 mm in size. However, it has an accelerated computing power of up to 21 million operations, consuming only 10–20 watts, allowing the user to simultaneously run multiple advanced neural networks and process data from multiple high-resolution sensors, which can help us perform gesture recognition faster.

The overall system architecture of the study is demonstrated in Figure 1. The BM201-PC3 mm wave radar collects point cloud data immediately when a user makes a gesture. These data are processed on the Jetson Xavier NX embedded evaluation board, which creates a real-time analysis of time-series results and recognition of the five periodic dynamic gestures we designed, as shown in Figure 2. To begin with, we trained the hand image recognition model using YOLOv7. Afterward, the model is integrated into the Jetson Xavier NX, and the Lepton 3.5 thermal imager is utilized to capture real-time hand image data. Jetson Xavier NX then conducts a real-time analysis of hand image movements, which produces time-series data. After analyzing the data obtained from both the millimeter wave radar and thermal imager, the resulting gesture is communicated to the user through audio feedback.

### 2.1. Image Processing of Hand Infrared Image

The temperature range for human palms is typically between 30 and 35 °C. In this study, two color conversion approaches are proposed: single-color and multi-color. In the single-color approach, the thermal imager detects the infrared radiation energy emitted by the object and converts it into temperature data. Pixels below 30 °C are filtered and appear colorless, while pixels above 30 °C are converted to red and displayed. However, this approach has some limitations. High body or room temperatures could cause multiple regions in the image to appear red, leading to unclear or non-existent hand image features, as shown in Figure 3. To address this issue, a multi-color conversion method is proposed. Firstly, the test was conducted in an indoor winter environment with a room temperature of approximately 22–26 °C. Observations showed that the measured hand temperature typically ranged between 30 and 36 °C during periods of minimal hand movement. However, when there was direct sunlight or a computer in the room, the thermal imager recorded temperatures above 36 °C. The multi-color conversion method is designed to reduce the effect of environmental factors that could obscure hand features, thereby improving the clarity of the hand image. When displaying the palm temperature, pixels below 30 °C are filtered out and remain black. Pixels with temperatures between 30 and 32 °C are displayed in red, while those between 32 and 34 °C are displayed in orange. Yellow pixels represent temperatures between 34 and 36 °C, and purple pixels represent temperatures above 36 °C. Details of the multi-color conversion are shown in Table 1.

### 2.2. Thermal Palm Image Detection Model

In this research, the YOLOv7 model is used to train the palm detection model. Two different datasets were compiled: one consisted of single-color converted images derived from a thermal imager, while the other consisted of multi-color converted images. The original image captured by the thermal imager was a jpg file with a resolution of 160 × 120 pixels, but for easier viewing and labeling, we scaled the image to a jpg file with a resolution of 400 × 300 pixels. Due to the similarity in palm images captured by the thermal imager, both datasets contain a total of 437 photographs taken by the same person. Before training, the objects were labeled using the labeling image labeling tool, with the designated classification categories including palm image and person, as shown in Figure 4. Data enhancement and dataset segmentation were performed using the Roboflow website. The dataset is split, with 95% allocated to training and 3% and 2% allocated to testing and validation, respectively. The palm model is then trained. Training of the YOLOv7 hand image recognition model was performed using Google Colab. Since the image training set is not a huge dataset, 80 iterations are sufficient for the loss function to finish convergence. We set a batch size of 8 for 80 iterations. The resulting palm image recognition model was then saved.

### 2.3. Gesture Point Cloud Data of mmWave

When millimeter wave radar detects a moving object, it generates point cloud information. This section describes how to use point cloud data from mmWave radar for neural network training, as illustrated in Figure 5. First, we need to collect the point cloud information for each gesture, followed by pre-processing to generate time-series feature data. Finally, these data are imported into the neural network for model training, resulting in the generation of the model file. Figure 6 shows the visual point cloud image of the mmWave radar detecting movement within the current range in the mmWave point cloud measurement screen. Specifically, Figure 6b shows the point cloud information resulting from rapid forward and backward hand movements. The box indicates the measurement range of the mmWave radar.

The acquired point cloud data include an elevation angle, azimuth angle, Doppler velocity, range (distance to radar), and SNR (signal-to-noise ratio). The elevation angle (ϕ), azimuth angle (*θ*), and range (R) data can be used to determine the x, y, and z positions of the point cloud. The definition of the mmWave coordinate axis is shown in Figure 7. The conversion of the three values—elevation angle, azimuth angle, and range—to Cartesian coordinates of the point cloud is shown in Equation (1) [35].
(1)$$\begin{array}{l} x=R*\cos\left(\phi \right)*\sin\left(\theta \right)\\ y=R*\cos\left(\phi \right)*\cos\left(\theta \right)\\ z=R*\sin(\phi ) \end{array}$$

### 2.4. Point Cloud Data Pre-Processing

In this section, we will discuss the pre-processing methods used for point cloud data obtained from the mmWave radar. The objective is to filter out any environmental noise, retain the hand-point cloud data, and extract its time-series signature information. The point cloud data go through a series of processing steps, including superposition, maximum speed limit, initial density-based spatial clustering of applications with noise (DBSCAN), registration, K-means, secondary DBSCAN, and extraction of time-series feature data. Afterward, the extracted time-series features are normalized before being imported into the neural network for training. To train the gesture recognition model, we first need to record the point cloud data of each gesture. A frame refers to the distribution of point clouds detected by the mmWave radar in a single instant. The mmWave radar is set to capture 200 frames in succession. The point cloud data collected from these 200 frames are then overlaid on the same array, resulting in 181 datasets. A detailed flowchart of this process is shown in Figure 8. As users perform gestures by waving their hands, the movement typically occurs at a relatively moderate speed. As a result, point clouds with speeds exceeding two meters per second are identified as potential noise points and then excluded. The removal process aims to improve data quality. The DBSCAN algorithm from the Scikit-Learn library is then applied to filter out any outliers, as these are likely to be environmental artifacts rather than the relevant hand and body point cloud information. After the first DBSCAN, the registered point cloud of the Counterclockwise gesture is shown in Figure 9. In the subsequent registration step, point clouds from different locations are aligned to a fixed reference point by rotation and translation. This alignment facilitates consistent data processing and increases recognition accuracy. To distinguish between hand and body point cloud information, the K-means clustering algorithm from the Scikit-Learn library is used, resulting in the segmentation of the point cloud data into two clusters: hand and body. A second DBSCAN is then performed to filter out any remaining body point cloud information. The final step involves the extraction of time-series signature data. Due to the dynamic and cyclic nature of the designed gestures, visual appearance alone is not sufficient for gesture identification. Therefore, time-series signature information is extracted from the pre-processed hand-point cloud data. Each set of 20 frames includes four eigenvalues, which include the normalized x, y, and z coordinate positions and the normalized velocity value.

### 2.5. mmWave Gesture Detection Model

During this research, three different types of neural networks were built using PyTorch: recurrent neural networks (RNNs), Long Short-Term Memory (LSTM), and Gated Recurrent Units (GRUs). The recurrent neural network [36] is different from the general feedforward neural network in that the message transfer in the feedforward neural network is only in one direction. However, in the recurrent neural network, the output value of the neuron at the current stage is sent back to the neural network and used as the input for the next stage or other neurons. In other words, the results of the current stage of message processing are retained in the neural network as a reference for the next stage of processing through message return. Because of this feature, the recurrent neural network has short-term memory and can find the temporal relationship in the data, which is widely used in natural language processing, handwriting recognition, time-series prediction, etc. Although the RNN has short-term memory, its disadvantage is that when the input data are too long, it is easy to generate gradient vanishing and gradient explosion. Compared with RNN, LSTM can handle long time-series data and solve the problem of gradient vanishing. However, due to the large size of its neural network model, the computation time is longer and data processing is more time-consuming. GRU is also a variant of RNN, but its structure is simpler than LSTM, which makes GRU faster in execution and computation. Therefore, we selected these three neural networks for our study. One person recorded 200 frames of point cloud data for each gesture, resulting in a total of 16 samples. The total number of point cloud data for five gestures is 16,000 frames. After pre-processing, each sample generated 181 time-series feature data. Consequently, each gesture produced 2896 time-series feature data after pre-processing, resulting in a total of 14,480 time-series feature data for the five gestures. After the model was trained, it was imported into the Jetson Xavier NX for gesture recognition. When the program is turned on, the mmWave radar will be in standby mode. If an object is detected within one meter, the program pauses. After collecting 20 frames of point cloud data, the speaker emitted a “stop” voice signal, and the collected point cloud data were saved. Jetson Xavier NX then processed the data and extracted time-series signature information, which was then fed into the neural network model. Gesture predictions were made, and the predicted gesture speech was played back.

## 3. Results

In this section, we will explain the results of the research questions, which are divided into three parts. These sections cover the training results for hand image recognition using a thermal imager and the training results for the gesture recognition model. Within gesture recognition, the discussion is further divided into two facets: the results obtained using mmWave radar alone and the results obtained using a hybrid approach combining both mmWave radar and thermal imager.

### 3.1. Thermal Imager Hand Image Recognition

Unlike an RGB camera, the thermal imager uses a special process to capture images. It filters out areas where the temperature falls below a certain threshold and performs color conversion for areas above this threshold. Figure 10 shows the two main types of training datasets for hand recognition. The first type, single-color conversion, identifies areas in the image with temperatures above 30 degrees Celsius and converts them to red while ignoring areas below this temperature. The second type is multi-color conversion, which converts the image into different colors based on the temperature zones above 30 degrees Celsius. Both training datasets underwent data enhancement. The dataset consists of 808 photographs, divided into 768 training set images, 23 validation set images, and 17 test set images. The training datasets were then subjected to 80 iterations. Figure 11 shows the loss function of the model after training, with notable differences in convergence between the single-color and multi-color training models.

We made several noteworthy observations during the experiment on hand-image recognition. It was discovered that the performance of the single-color conversion process was affected by both ambient and body temperatures. Specifically, when the temperature increases slightly or the body temperature rises, the area of color in the image merges with the background, as illustrated in Figure 12a. This blending of colors in the image led to decreased recognition capabilities since the features of the hand image could not be accurately distinguished. On the other hand, the multi-color conversion was also tested, and the result is shown in Figure 12b. During the test, ambient and body temperatures were elevated, causing the colored area in the image to dominate most of the screen. However, the multi-color conversion method prevents the blending of hand image features with the background. This resulted in a significant improvement in recognition rate and accuracy, even when the background is intricate. Table 2 provides a comparative analysis between the single-color and multi-color conversions, demonstrating that multi-color conversion achieves superior accuracy compared to its single-color counterpart.

### 3.2. Gesture Recognition Using mmWave Radar

The point cloud data for the hand gestures undergoes pre-processing to isolate the hand’s point cloud data. Figure 13 shows the sequential pre-processing results for the counterclockwise gesture, including superposition, maximum speed limit, first DBSCAN, alignment of hand and body, K-means separation, and second DBSCAN. Subsequently, the time-series data of the point cloud is extracted for training purposes. To enhance the efficiency of training and data processing, we make use of normalization techniques. This helps us standardize the data and make them more consistent, which in turn leads to better accuracy and faster processing times. The MinMaxScaler is utilized for the (x, y, z) coordinates of the time signature data, thus scaling the data to a range of 0 to 1. Furthermore, MaxAbsScaler is applied to the average speed of the time signature data, scaling the data to the range of −1 to 1. For clarity, we extract six frames from the output to observe the center of mass changes in the point cloud, as demonstrated in Figure 14. The red dot represents the current center of mass, while the blue dot represents the previous one. It is apparent that the center of mass for the clockwise gesture shifts in a clockwise circle while the counterclockwise gesture shifts in a counterclockwise circle. In the same way, the right gesture shifts horizontally to the right, and the left gesture shifts horizontally to the left. Lastly, the punch gesture shifts vertically up and down.

This study analyzed the results of mmWave radar’s hand image recognition using 14,480 sample data from five different gestures. The data were separated into three sets randomly, with 60% for training and 20% each for validation and testing. This study trained the GRU, LSTM, and RNN models using these sets over forty iterations. Figure 15 shows the confusion matrix for all models, indicating prediction accuracies of 99.51%, 99.37%, and 81.11% for GRU, LSTM, and RNN, respectively. GRU performed better than the other models. In terms of the prediction time, GRU took 0.462 ms, LSTM took 0.483 ms, and RNN took 0.461 ms, with RNN being the fastest. However, its accuracy did not match that of GRU and LSTM. Table 3 illustrates the accuracy rates of the three models during the gesture recognition test. In our testing, we waved each gesture 10 times to determine their correct identification. This highlighted the superiority of the GRU model over LSTM and RNN in terms of accuracy and efficiency.

### 3.3. Gesture Recognition Using mmWave Radar with a Thermal Imager

Continuing from the previous section, this section provides further details on the gesture recognition technique that combines an mmWave radar and a thermal imager. In addition to the millimeter wave point cloud data, we will also extract the normalized time signature data and the coordinate timing change of the thermal imager for gesture recognition. During the gesture test, the thermal imager uses YOLOv7 to recognize the hand image and record the timing changes in its coordinates. Figure 16 highlights the resulting coordinate changes for the five gestures. The mmWave radar captured 20 frames of point cloud data during gesture recognition. Nonetheless, the thermal imager employed for YOLOv7 hand recognition operates at a slower execution speed. This study revealed that the duration necessary for the mmWave radar to acquire 20 frames was equivalent to the time required for the thermal imager to process 12 frames of hand image recognition. In some cases, the thermal imager may not identify the hand image in a given frame, resulting in less than 12 frames of recorded time-series data. To incorporate the thermal imager’s coordinate timing changes into the mmWave time signature data for gesture model training, interpolation is carried out on the thermal imager’s coordinate timing change curve. This expands the data to 20 frames without modifying the waveform. Subsequently, the data are scaled using MinMaxScaler nine times, and the outcomes are fused into 200 frames of time-series data. The model for gesture recognition, which uses both a thermal imager and mmWave, is an improvement over the one that only relies on mmWave. During training, the model utilizes the average mmWave velocity, the time-series variation of thermal imager coordinates, and mmWave time-series signature data as inputs. We trained the model using GRU, LSTM, and RNN for 40 iterations. A total of 14,480 data were randomly divided into three sets: 60% for training, 20% for validation, and 20% for testing for the five gestures. The model’s performance was evaluated using a confusion matrix in Figure 17, which shows that the GRU and LSTM models achieved prediction accuracies of 100%, while the RNN model achieved an accuracy of 98.14%. Table 4 outlines the results of real gesture recognition tests, comparing the accuracy of mmWave technology alone with the combination of mmWave radar and a thermal imager. The analysis reveals a significant improvement in precision when the two technologies are combined. Among these, the results show that clockwise and counterclockwise gestures have a higher accuracy rate than the other three. During the analysis of gesture recognition, we measured the time required to recognize a gesture in the model training test and the time taken to import it into Jetson Xavier NX for actual recognition. This can be observed in Figure 18. This figure shows that the time needed for real-time recognition in Jetson Xavier NX is longer than that observed during the model validation test in Google Colab. This is because the embedded system has to process multiple programs at the same time, resulting in increased memory usage and reduced performance.

## 4. Discussion

Our research indicates that by combining millimeter waves and thermal imaging, we can enhance the precision of hand gesture recognition. These results support our hypothesis that the thermal imager can be used to detect hand images while maintaining privacy. However, it is important to acknowledge that the findings of this study are subject to some limitations. For instance, the thermal imager may not be able to capture hand images accurately in situations where the ambient temperature is too high or when the body temperature is close to the ambient temperature, which may limit the universality of our results. Nevertheless, this study has several advantages. These include the inclusion of the coordinate timing variations of the thermal imager in the millimeter wave time signature data. Millimeter waves can detect objects with relative velocity variations and produce point cloud data, while the thermal imager is particularly good at detecting lateral motion. This helped us identify gesture locations and variations, which improved the accuracy of our results. In terms of gesture design, we found that clockwise and left gestures and counterclockwise and right gestures have similar swing trajectories, which makes it easy to misjudge gestures. Future research could further analyze gesture design to clarify these issues.

## 5. Conclusions

In this study, a large-motion gesture recognition system is developed using deep learning techniques that integrate millimeter wave radar with a thermal imager. The palm image information is captured using an infrared thermal imager, and the coordinate movement changes in the palm on the screen are recorded at the same time. The point cloud data from the millimeter wave radar, including three-axis coordinates and velocity, is integrated and pre-processed to produce time-series data. These data are processed via a neural network for recognizing gestures, and real-time recognition is achieved through the use of the Jetson Xavier NX embedded evaluation board. The results demonstrate that the accuracy of gesture recognition with the combination of millimeter wave radar and the thermal imager is significantly better than with millimeter wave radar alone. As before, the model trained with a Gated Recurrent Unit outperforms the Long Short-Term Memory and recurrent neural network models in gesture recognition tasks. This study advances the development of multimodal gesture recognition systems such as gaming, somatosensory applications, virtual reality, etc., highlighting the potential for higher accuracy and performance through the integration of various sensing technologies and deep learning approaches.

## Figures and Tables

**Figure 1 sensors-24-00581-f001:**
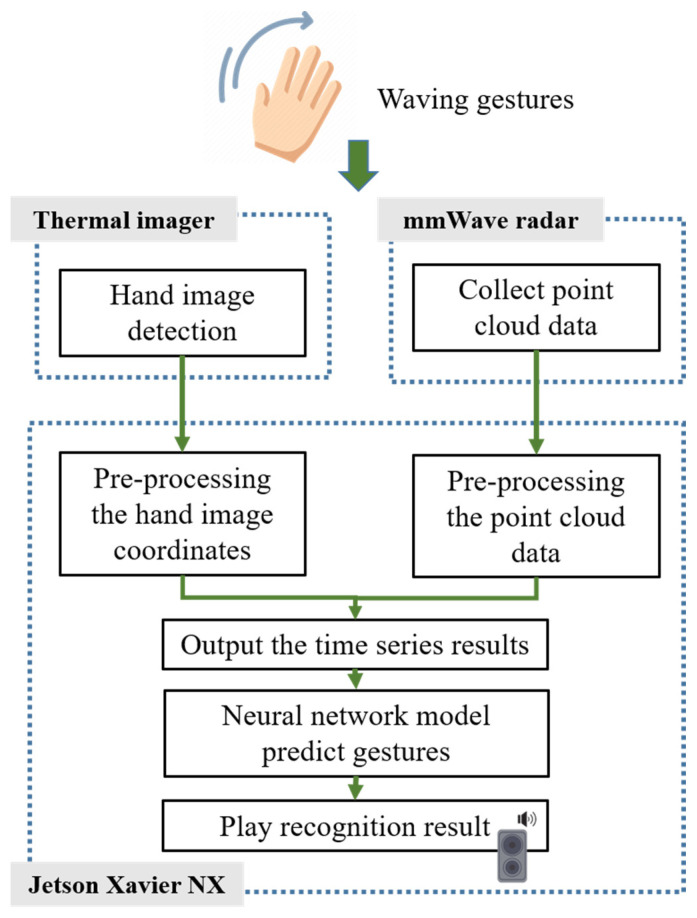
The system architecture.

**Figure 2 sensors-24-00581-f002:**
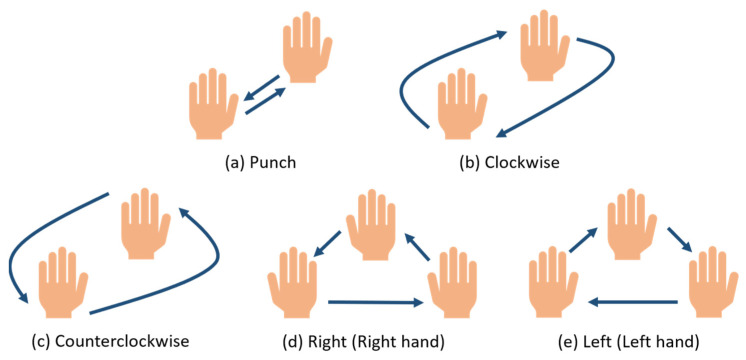
The five periodic dynamic gestures.

**Figure 3 sensors-24-00581-f003:**
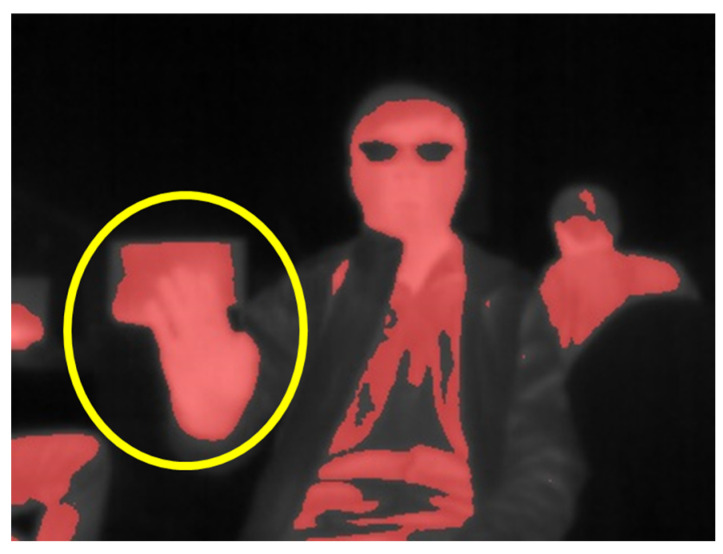
The thermal image at a higher ambient temperature.

**Figure 4 sensors-24-00581-f004:**
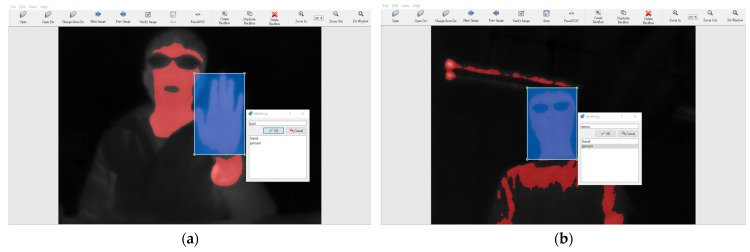
The object labeling of (**a**) palm and (**b**) person.

**Figure 5 sensors-24-00581-f005:**
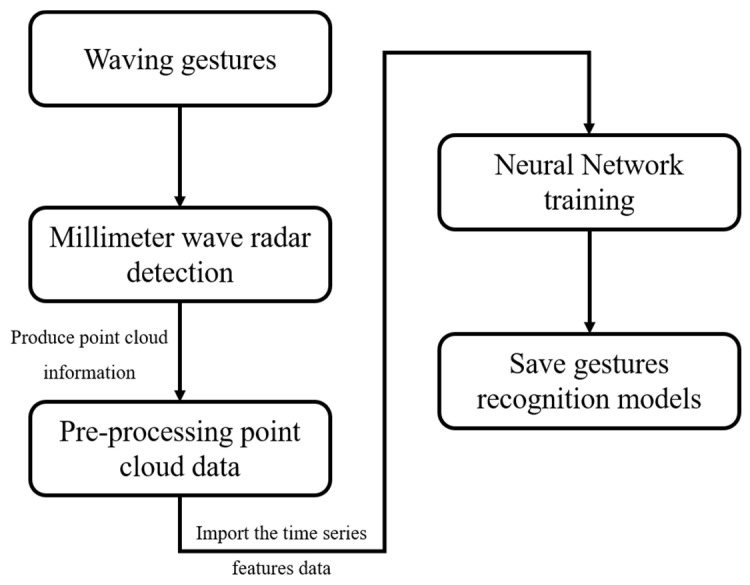
The procedure for training a gesture recognition model.

**Figure 6 sensors-24-00581-f006:**
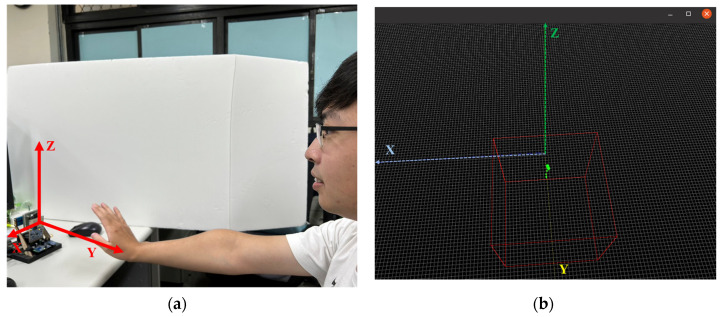
(**a**) Make gesture. (**b**) The mmWave point cloud screen.

**Figure 7 sensors-24-00581-f007:**
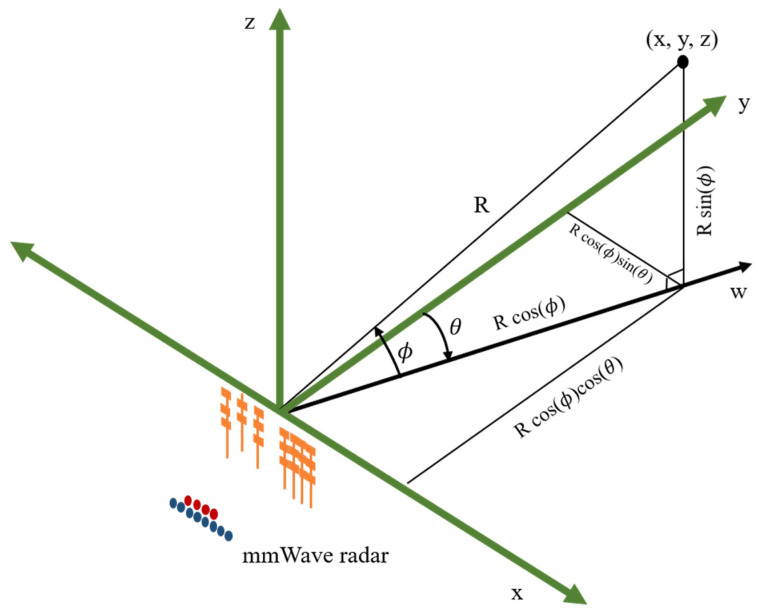
The definition of BM201-PC3 mmWave coordinate axis [35].

**Figure 8 sensors-24-00581-f008:**
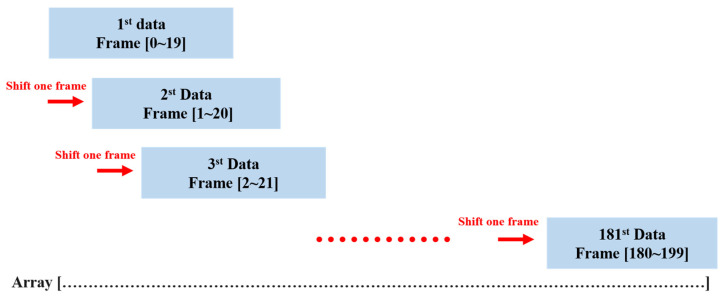
The flow of superimposed point cloud data on the same array.

**Figure 9 sensors-24-00581-f009:**
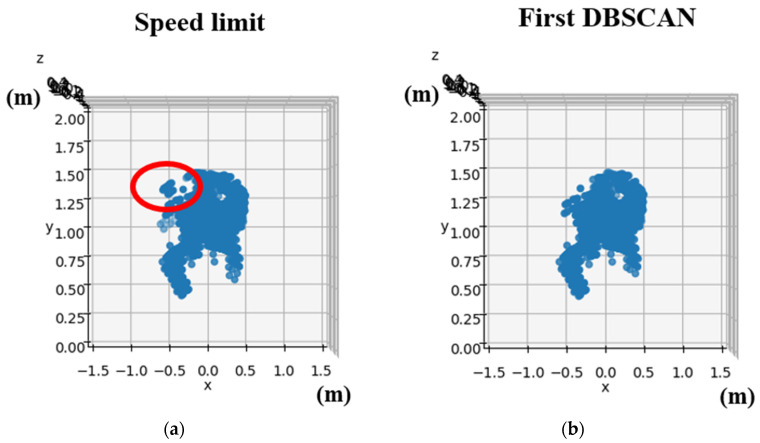
The pre-processing point cloud data of counterclockwise gesture: (**a**) Speed Limit. (**b**) First DBSCAN.

**Figure 10 sensors-24-00581-f010:**
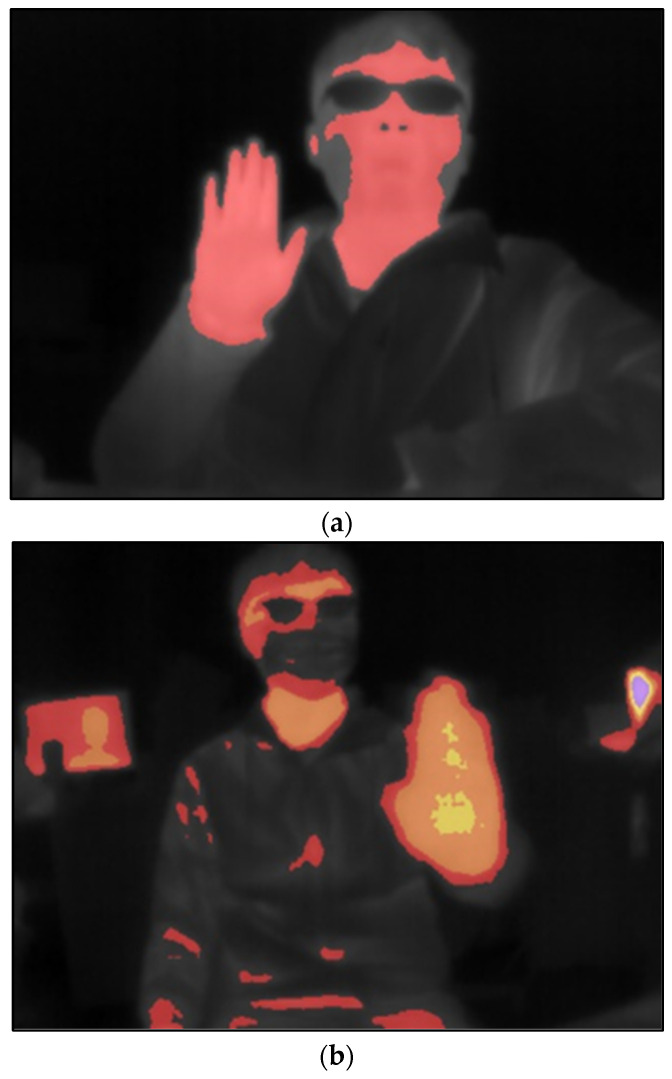
(**a**) Single-color conversion. (**b**) Multi-color conversion.

**Figure 11 sensors-24-00581-f011:**
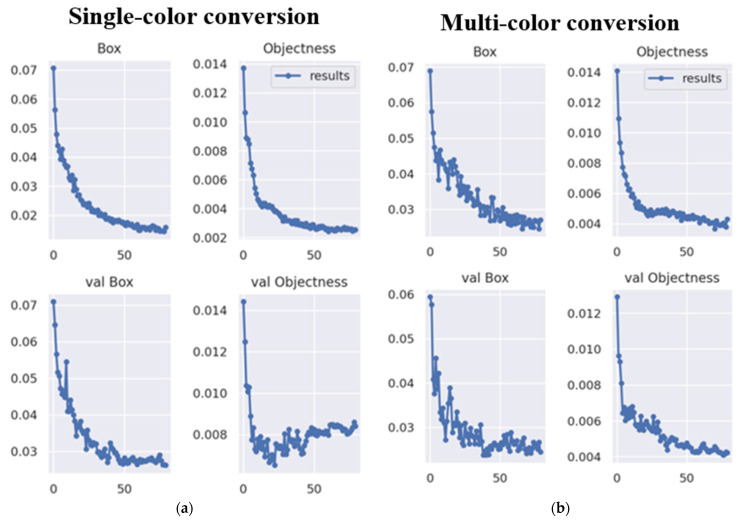
The loss function of (**a**) single-color and (**b**) multi-color.

**Figure 12 sensors-24-00581-f012:**
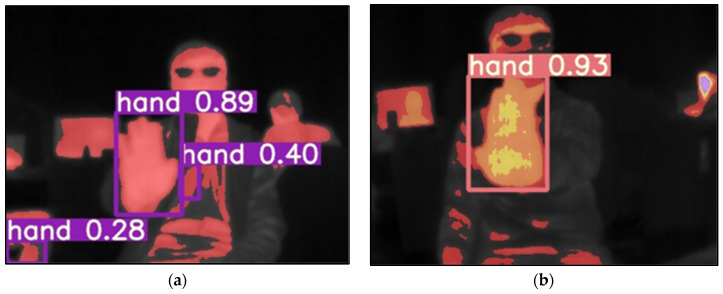
The hand-image recognition of (**a**) single-color and (**b**) multi-color conversion.

**Figure 13 sensors-24-00581-f013:**
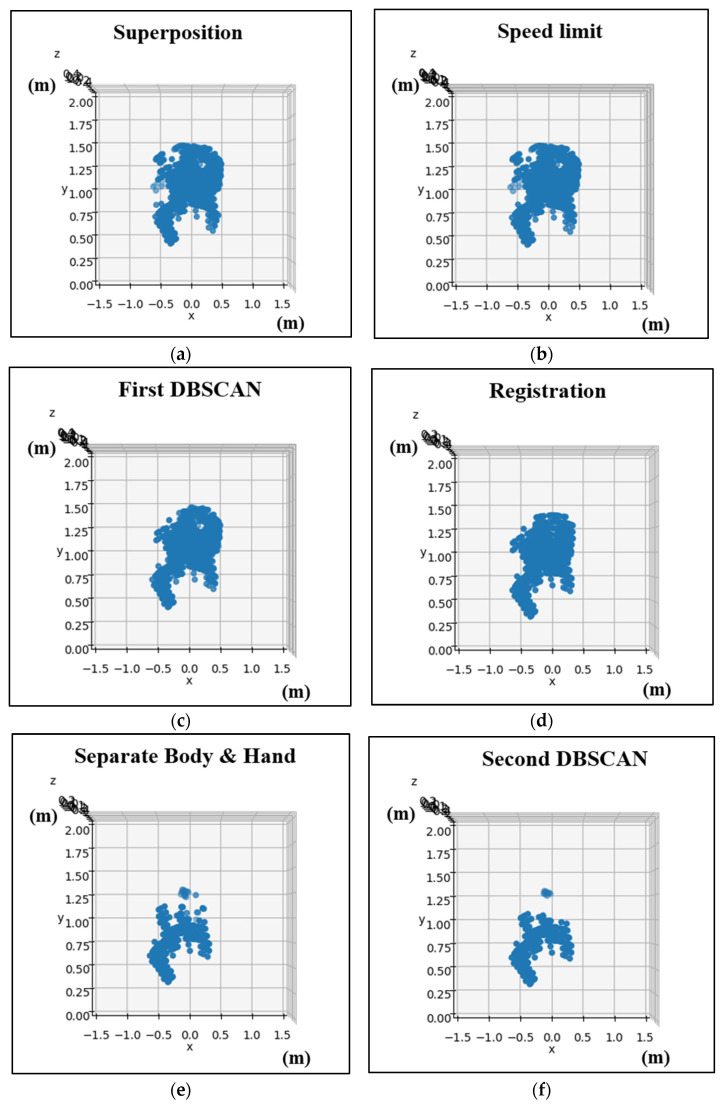
The pre-processing results of the clockwise gesture: (**a**) superposition; (**b**) speed limit; (**c**) first DBSCAN; (**d**) registration; (**e**) separate body and hand; and (**f**) second DBSCAN.

**Figure 14 sensors-24-00581-f014:**
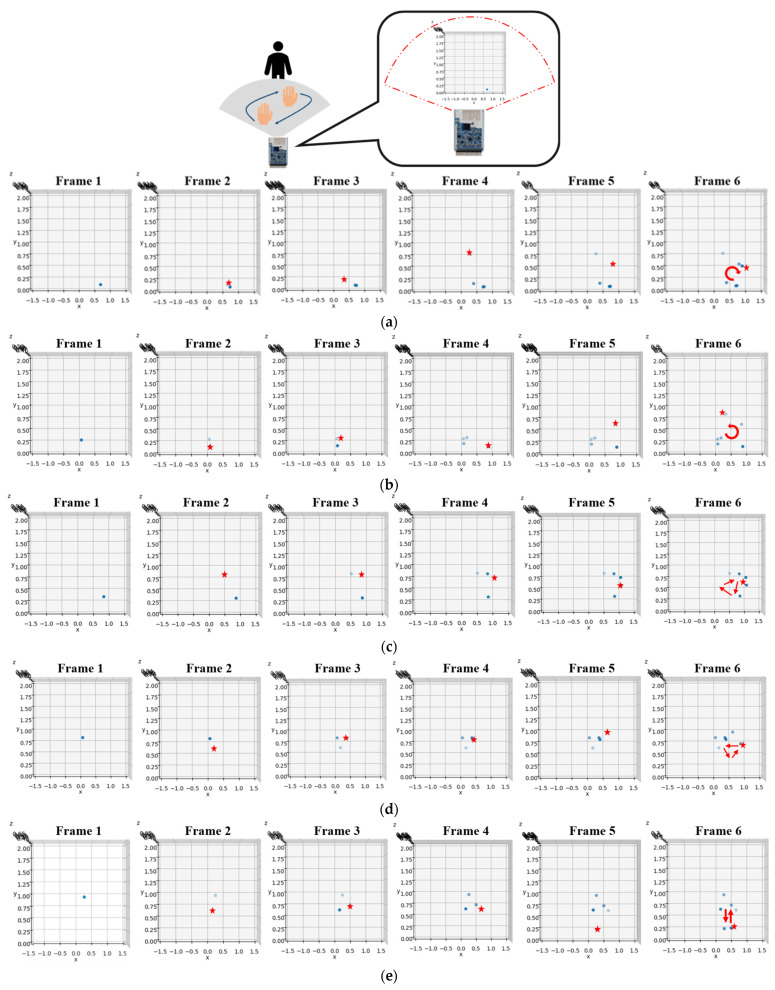
The mass of the point cloud for the gesture: (**a**) clockwise; (**b**) counterclockwise; (**c**) left; (**d**) right; and (**e**) punch, where the red star represents the current center of mass.

**Figure 15 sensors-24-00581-f015:**
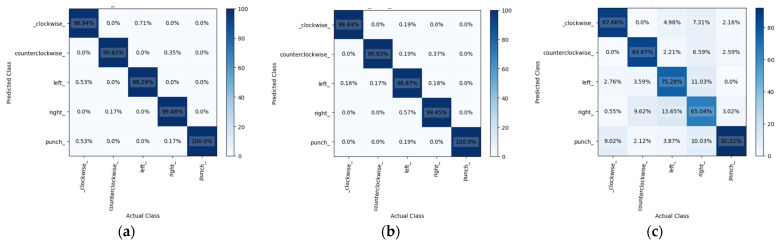
The confusion matrix of (**a**) GRU, (**b**) LSTM, and (**c**) RNN using mmWave radar.

**Figure 16 sensors-24-00581-f016:**
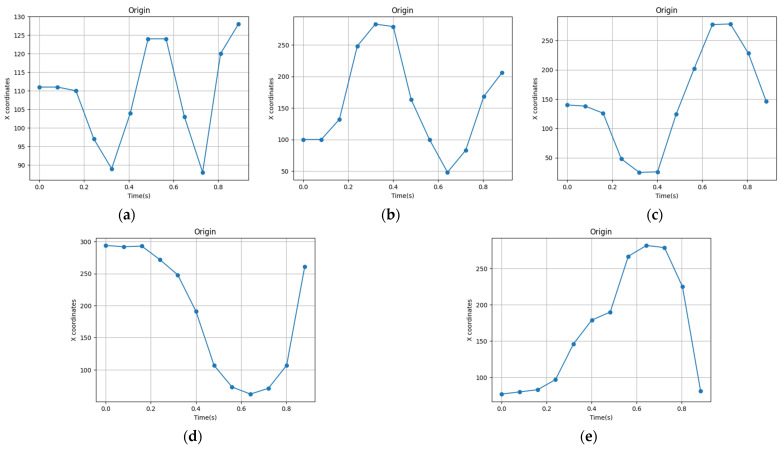
The coordinates change for the five gestures: (**a**) punch, (**b**) clockwise, (**c**) counterclockwise, (**d**) left, and (**e**) right.

**Figure 17 sensors-24-00581-f017:**
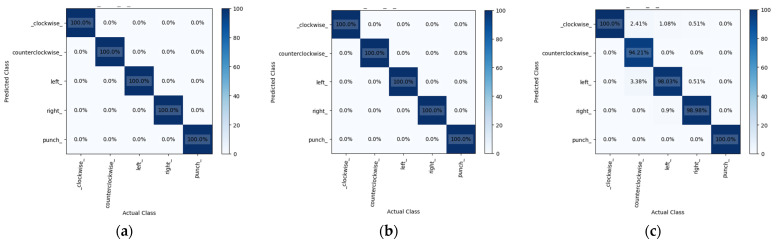
The confusion matrix of (**a**) GRU, (**b**) LSTM, and (**c**) RNN using mmWave radar with thermal imager.

**Figure 18 sensors-24-00581-f018:**
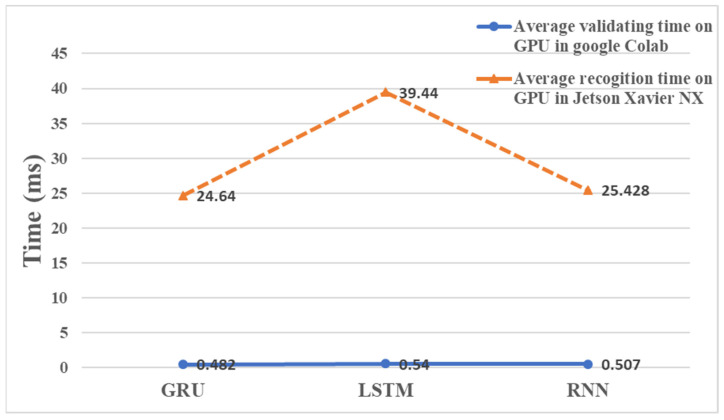
The gesture recognition time of mmWave combined with a thermal imager.

**Table 1 sensors-24-00581-t001:** Thermal imager temperature versus multi-color conversion.

Multi-Color Conversion	Degrees Celsius (°C)
Filtered out	<30
Red	30~32
Orange	32~34
Yellow	34~36
Purple	>36

**Table 2 sensors-24-00581-t002:** The accuracy of single-color and multi-color conversion.

Testing Times	Single-Color	Multi-Color
1	0.8	0.94
2	0.91	0.93
3	0.91	0.94
4	0.91	0.93
5	0.92	0.9
6	0.94	0.93
7	0.93	0.94
8	0.8	0.92
9	0.92	0.94
10	0.95	0.96
**Average**	**0.899**	**0.933**

**Table 3 sensors-24-00581-t003:** The accuracy of the three models using mmWave radar.

	GRU	LSTM	RNN
Punch	70%	70%	50%
Clockwise	80%	80%	60%
Counterclockwise	80%	70%	50%
Left	70%	70%	40%
Right	70%	60%	50%
**Average**	**74%**	70%	50%

**Table 4 sensors-24-00581-t004:** The accuracy of the three models using mmWave radar with thermal imager.

	mmWave	mmWave and Thermal Imager	mmWave	mmWave and Thermal Imager	mmWave	mmWave and Thermal Imager
	GRU		LSTM		RNN	
Punch	70%	80%	70%	90%	50%	70%
Clockwise	80%	90%	80%	80%	60%	80%
Counterclockwise	80%	90%	70%	80%	50%	60%
Left	70%	80%	70%	80%	40%	60%
Right	70%	80%	60%	70%	50%	50%
**Average**	74%	**84%**	70%	**80%**	50%	**64%**

## Data Availability

Data are contained within the article.

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
