# Peer review of "The Implementation of a Gesture Recognition System with a Millimeter Wave and Thermal Imager"

_sensors, 2024, doi:10.3390/s24020581_

Round 1

Reviewer 1 Report

Comments and Suggestions for Authors

The Title: Implementation of a gesture recognition system with millimeter wave and thermal imager.

This study introduces an innovative gesture recognition system for large movements that uses a combination of millimeter- wave radar and thermal imager, together with deep learning approaches to improve its accuracy. The authors reflected their works faithfully and presented well organization. I believe it can be accepted after addressing the following important issues:

The comments:

1-The introduction section is weak and requires adding some recent related works and providing a discussion of the existing gaps leading to a proposal for the current work. It is recommended that a review of relevant works and inclusion of references be conducted during the period 2020-2023.

2-There are some grammatical and linguistic errors that need to be corrected. For instance:

-In line 35, it is stated that “The first method involves….”. Therefore, it is required to mention the second method obviously.

-In line 73, it is stated that “The study introduces …”. It is better to write “This study introduces …

- The organization of the paper stats in line 77, should be started as a new paragraph.

- Previous works in the introduction section have been mentioned in different tenses. Please standardize the tenses appropriately.

- In line 215, it is stated that “In this chapter, we will…”. Please replace the word chapter.

Please check such issues for the whole manuscript.

3-It is recommended to highlight the contribution of the proposed work more clearly.

4- A comparative analysis has been made between the single-color and multi-color conversion. However, state-of-the-art algorithms should be included with comparison with existing work to show how superior the proposed method is to existing algorithms and this point should be mentioned in the abstract.

5- It is recommended to start section 2 with a brief description of the materials used in the proposed work to provide a clear vision to the reader.

6- Justification is required as to why dynamic gestures should be considered in the proposed work instead of static gestures.

7- The dataset for Section 2.2 was taken by the same person. Based on what? Why was the standard data set not used?

8- Any figures, data or equations should be cited with a reliable reference if they are taken from another source.

9- The resolution of Figure 9 should be enhanced. Please check such issues for the other figures.

10- It is better to add some points for future trends.

Comments on the Quality of English Language

Minor editing of English language required

Author Response

Manuscript: Sensors ID -2772780

Response to Reviewers

Dear Dr./Mr./Ms.: 

Thank you for inviting us to submit a revised draft of our manuscript entitled, " Implementation of a gesture recognition system with millimeter wave and thermal imager" to Sensors. We also appreciate the time and effort you and each of the reviewers have dedicated to providing insightful feedback on ways to strengthen our paper. Thus, it is with great pleasure that we resubmit our article for further consideration. We have incorporated most of the suggestions of reviewers. Please see below, in blue, for a point-by-point response to the reviewers’ comments and questions.

REVIEWER 1 COMMENTS:

This study introduces an innovative gesture recognition system for large movements that uses a combination of millimeter-wave radar and thermal imager, together with deep learning approaches to improve its accuracy. The authors reflected their works faithfully and presented well organization. I believe it can be accepted after addressing the following important issues:

Author response: Thank you!

  1. The introduction section is weak and requires adding some recent related works and providing a discussion of the existing gaps leading to a proposal for the current work. It is recommended that a review of relevant works and inclusion of references be conducted during the period 2020-2023.

RESPONSE: Thank you for pointing this out. The reviewer is correct, and we have added more recent and relevant literature to section 1 of the manuscript.

  1. There are some grammatical and linguistic errors that need to be corrected. For instance:

-In line 35, it is stated that “The first method involves….”. Therefore, it is required to mention the second method obviously.

-In line 73, it is stated that “The study introduces …”. It is better to write “This study introduces …”

- The organization of the paper stats in line 77, should be started as a new paragraph.

- Previous works in the introduction section have been mentioned in different tenses. Please standardize the tenses appropriately.

- In line 215, it is stated that “In this chapter, we will…”. Please replace the word chapter.

Please check such issues for the whole manuscript.

RESPONSE: Thank you for providing these insights. We agree with you and have revised the sentences and terms incorporated in this suggestion throughout our paper.

  1. It is recommended to highlight the contribution of the proposed work more clearly.

RESPONSE: We think this is an excellent suggestion. Thank you for this suggestion. We agree with you and have incorporated this suggestion throughout our paper. We have redrafted the introduction section (lines 103-106) to establish a clearer focus.

  1. A comparative analysis has been made between the single-color and multi-color conversion. However, state-of-the-art algorithms should be included with comparison with existing work to show how superior the proposed method is to existing algorithms and this point should be mentioned in the abstract.

RESPONSE: Thank you for pointing this out. We have revised the abstract (lines 13-14) to highlight the contribution.

  1. It is recommended to start section 2 with a brief description of the materials used in the proposed work to provide a clear vision to the reader.

RESPONSE: Thank you for providing these insights. We have added the suggested content to the manuscript on section 2 of page 3 where the change can be found in the revised manuscript.

  1. Justification is required as to why dynamic gestures should be considered in the proposed work instead of static gestures.

RESPONSE: Thank you for providing these insights. We have added the suggested content to the manuscript in section 1 (lines 61-65) of page 2 to establish a clearer focus.

  1. The dataset for Section 2.2 was taken by the same person. Based on what? Why was the standard data set not used?

RESPONSE: Thank you for providing these insights. You have raised an important question. We have added the suggested content to the manuscript in section 2.2 (lines 177-178) of page 5 to establish a clearer focus.

  1. Any figures, data or equations should be cited with a reliable reference if they are taken from another source.

RESPONSE: Thank you for your suggestion. We have added the reference to cite the figure and equation.

  1. The resolution of Figure 9 should be enhanced. Please check such issues for the other figures.

RESPONSE: Thank you for pointing this out. We have changed Figure 9 to a high-resolution image, and we have also checked the images throughout the manuscript and replaced them.

  1. It is better to add some points for future trends.

Thank you for your suggestion. We have added the suggested content to the manuscript in section 4 (line 423) of page 17 to establish a clearer focus.

CONCLUDING REMARKS: Again, thank you for giving us the opportunity to strengthen our manuscript with your valuable comments and queries. We have worked hard to incorporate your feedback and hope that these revisions persuade you to accept our submission.

Sincerely,

Yu-Ping Liao

Name: Yu-Ping Liao

Institute: Chung Yuan Christian University, Taiwan

Address: No. 200, Zhongbei Rd., Zhongli Dist., Taoyuan, 320

E-mail: lyp@cycu.org.tw

Reviewer 2 Report

Comments and Suggestions for Authors

Some comments:

1. The authors introduce designing a gesture recognition system with millimeter wave and thermal imager, but it isn't easy to find how the recognition using the imager is improved with the help of mmW radar.

2. Can the authors specify what kind of gestures can be recognized as the point cloud data shows the gesture at meters-level?

3. Do the authors consider some fusion methods to improve the recognition?

Author Response

Manuscript: Sensors ID -2772780

Response to Reviewers

Dear Dr./Mr./Ms.: 

Thank you for inviting us to submit a revised draft of our manuscript entitled, " Implementation of a gesture recognition system with millimeter wave and thermal imager" to Sensors. We also appreciate the time and effort you and each of the reviewers have dedicated to providing insightful feedback on ways to strengthen our paper. Thus, it is with great pleasure that we resubmit our article for further consideration. We have incorporated most of the suggestions of reviewers. Please see below, in blue, for a point-by-point response to the reviewers’ comments and questions.

REVIEWER 2 COMMENTS:

  1. The authors introduce designing a gesture recognition system with millimeter wave and thermal imager, but it isn't easy to find how the recognition using the imager is improved with the help of mmW radar.

RESPONSE: Thank you for pointing this out. We have revised the table 4 to establish a clearer focus.

  1. Can the authors specify what kind of gestures can be recognized as the point cloud data shows the gesture at meters-level?

RESPONSE: Thank you for providing these insights. However, since the point cloud data of each gesture will be too similar, we judge the gestures by the center of mass of the point cloud for easy observation, as shown in Figure 14.

  1. Do the authors consider some fusion methods to improve the recognition?

RESPONSE:Thank you for pointing this out. Although the sampling data is done in separate modules. But, in the training of the neural network model, we use the data from both sensors for neural network inference, so the fusion method has been used to improve the recognition rate.

CONCLUDING REMARKS: Again, thank you for giving us the opportunity to strengthen our manuscript with your valuable comments and queries. We have worked hard to incorporate your feedback and hope that these revisions persuade you to accept our submission.

Sincerely,

Yu-Ping Liao

Name: Yu-Ping Liao

Institute: Chung Yuan Christian University, Taiwan

Address: No. 200, Zhongbei Rd., Zhongli Dist., Taoyuan, 320

E-mail: lyp@cycu.org.tw

Reviewer 3 Report

Comments and Suggestions for Authors

This paper presents a deep learning-based method with the data captured by millimeter-wave radar with a thermal imager for large-motion gesture recognition. The authors made efforts to demonstrate that incorporating a thermal imager with millimeter-wave radar could effectively improve gesture recognition accuracy.

Strong points:

This paper presents a method that uses a deep neural network to process the point cloud information captured by mmWave radar and adopts thermal imaging and pal recognition to resolve the suboptimal recognition results with machine vision under low light conditions and complex backgrounds.

Weak points.

1. This paper is written in a way more like a technical report than a research paper.

2. The paper is not well written and organized, and not easy to follow. 

3. The authors claimed that most current gesture recognition technologies rely on machine vision methods which can lead to suboptimal recognition results, especially in situations where the camera is 10 operating in low light conditions or encounters complex backgrounds. However, there is a lack of comparison between the proposed method with SOTA gesture recognition methods under the same conditions. 

Author Response

Manuscript: Sensors ID -2772780

Response to Reviewers

Dear Dr./Mr./Ms.: 

Thank you for inviting us to submit a revised draft of our manuscript entitled, " Implementation of a gesture recognition system with millimeter wave and thermal imager" to Sensors. We also appreciate the time and effort you and each of the reviewers have dedicated to providing insightful feedback on ways to strengthen our paper. Thus, it is with great pleasure that we resubmit our article for further consideration. We have incorporated most of the suggestions of reviewers. Please see below, in blue, for a point-by-point response to the reviewers’ comments and questions.

REVIEWER 3 COMMENTS:

This paper presents a deep learning-based method with the data captured by millimeter-wave radar with a thermal imager for large-motion gesture recognition. The authors made efforts to demonstrate that incorporating a thermal imager with millimeter-wave radar could effectively improve gesture recognition accuracy.

Strong points:

This paper presents a method that uses a deep neural network to process the point cloud information captured by mmWave radar and adopts thermal imaging and pal recognition to resolve the suboptimal recognition results with machine vision under low light conditions and complex backgrounds.

Author response: Thank you!

Weak points.

  1. This paper is written in a way more like a technical report than a research paper.

RESPONSE: Thank you for your suggestion. We agree with you and have incorporated this suggestion throughout our paper. We have revised some sentences in the manuscript to describe our views more clearly.

  1. The paper is not well written and organized, and not easy to follow.

RESPONSE: Thank you for your suggestion. We agree with you and have incorporated this suggestion throughout our paper. We have revised some sentences in the manuscript to describe our views more clearly.

  1. The authors claimed that most current gesture recognition technologies rely on machine vision methods which can lead to suboptimal recognition results, especially in situations where the camera is 10 operating in low light conditions or encounters complex backgrounds. However, there is a lack of comparison between the proposed method with SOTA gesture recognition methods under the same conditions.

RESPONSE: Thank you for providing these insights. You have raised an important point, however, this point may be outside the scope of our paper because we use millimeter wave and thermal imager to recognize gestures unlike RGB camera so the light source doesn't affect us.

CONCLUDING REMARKS: Again, thank you for giving us the opportunity to strengthen our manuscript with your valuable comments and queries. We have worked hard to incorporate your feedback and hope that these revisions persuade you to accept our submission.

Sincerely,

Yu-Ping Liao

Name: Yu-Ping Liao

Institute: Chung Yuan Christian University, Taiwan

Address: No. 200, Zhongbei Rd., Zhongli Dist., Taoyuan, 320

E-mail: lyp@cycu.org.tw

Reviewer 4 Report

Comments and Suggestions for Authors

This paper uses a combination of millimeter-wave radar and thermal imaging to collect data, and trains a hand detection model using the YOLOv7 network to track and monitor hand movements on a screen, achieving recognition rates of over 80%. The study only uses the original methods and YOLOv7 and time-series models without improving or enhancing the algorithms, and does not compare the performance with other mainstream object detection models. The language in the abstract and introduction is not concise enough, and I believe that articles in the field of computer science should use more language to describe algorithm performance improvements, while camera theory should be summarized more succinctly. Overall, further refinement of the article is needed. In conclusion, I did not find this research to make a valuable contribution to the literature or science

Comments for the Author:

l  It is suggested to discuss the "restraints of two-dimensional image capture" in more detail in the introduction.

l  The introduction claims that millimeter-wave radar can eliminate the influence of occlusion and foggy conditions, but these conditions were not tested or validated in this study. Therefore, the superiority of this method is not demonstrated.

l  The choice of YOLOv7 network over other object detection networks is not explained in the introduction, and the logical flow of the introduction is weak.

l  Please provide information about the resolution, image format, and other details of the captured images in line 125 to provide readers with a deeper understanding of the study.

l  How were the training hyperparameters for YOLOv7 selected? Why were 80 iterations chosen? The description of the YOLOv7 training process is too brief.

l  In "2.2. Thermal Hand Image Detection Model" (line 125), how many gesture categories are there? The discussion of annotated gesture categories is lacking. It is recommended to showcase more images of different gesture categories to enhance understanding of the study.

l  In "2.5. mmWave Gesture Detection Model" (line 199), why were RNN, LSTM, and GRU chosen? What are their advantages and disadvantages? These aspects are not discussed in the text.

l  The description of the deep learning gesture recognition part is limited, and there is not much discussion on the experimental section or performance comparison with other mainstream models.

l  The conclusion lacks logical rigor and requires further modification.

Based on the reasons mentioned above, I will have to reject this paper.

Comments on the Quality of English Language

 Minor editing of English language required

Author Response

Manuscript: Sensors ID -2772780

Response to Reviewers

Dear Dr./Mr./Ms.: 

Thank you for inviting us to submit a revised draft of our manuscript entitled, " Implementation of a gesture recognition system with millimeter wave and thermal imager" to Sensors. We also appreciate the time and effort you and each of the reviewers have dedicated to providing insightful feedback on ways to strengthen our paper. Thus, it is with great pleasure that we resubmit our article for further consideration. We have incorporated most of the suggestions of reviewers. Please see below, in blue, for a point-by-point response to the reviewers’ comments and questions.

REVIEWER 4 COMMENTS:

This paper uses a combination of millimeter-wave radar and thermal imaging to collect data, and trains a hand detection model using the YOLOv7 network to track and monitor hand movements on a screen, achieving recognition rates of over 80%. The study only uses the original methods and YOLOv7 and time-series models without improving or enhancing the algorithms, and does not compare the performance with other mainstream object detection models. The language in the abstract and introduction is not concise enough, and I believe that articles in the field of computer science should use more language to describe algorithm performance improvements, while camera theory should be summarized more succinctly. Overall, further refinement of the article is needed. In conclusion, I did not find this research to make a valuable contribution to the literature or science

Comments for the Author:

  1. It is suggested to discuss the "restraints of two-dimensional image capture" in more detail in the introduction.

RESPONSE: Thank you for pointing this out. The reviewer is correct, and we have added more recent and relevant literature to section 1 (lines 48-50) of the manuscript.

  1. The introduction claims that millimeter-wave radar can eliminate the influence of occlusion and foggy conditions, but these conditions were not tested or validated in this study. Therefore, the superiority of this method is not demonstrated.

RESPONSE: Thank you for pointing this out. The reviewer is correct, and we have added more recent and relevant literature to section 1 (lines 74-76) of the manuscript.

  1. The choice of YOLOv7 network over other object detection networks is not explained in the introduction, and the logical flow of the introduction is weak.

RESPONSE: Thank you for providing these insights. The reviewer is correct, and we have added more recent and relevant literature to section 1 (lines 33-37) of the manuscript.

  1. Please provide information about the resolution, image format, and other details of the captured images in line 125 to provide readers with a deeper understanding of the study.

RESPONSE: Thank you for your suggestion. We have added the suggested content to the manuscript in section 2.2 of page 5 where the change can be found in the revised manuscript.

  1. How were the training hyperparameters for YOLOv7 selected? Why were 80 iterations chosen? The description of the YOLOv7 training process is too brief.

RESPONSE: Thank you for pointing this out. We have added the suggested content to the manuscript in section 2.2 (lines 184-186) where the change can be found in the revised manuscript. We hope these revisions provide a more balanced discussion.

  1. In "2.2. Thermal Hand Image Detection Model" (line 125), how many gesture categories are there? The discussion of annotated gesture categories is lacking. It is recommended to showcase more images of different gesture categories to enhance understanding of the study.

RESPONSE: Thank you for pointing this out. We have replaced the text hand with the palm to clarify that this section is an illustration of how the thermal imager recognizes the palm.

  1. In "2.5. mmWave Gesture Detection Model" (line 199), why were RNN, LSTM, and GRU chosen? What are their advantages and disadvantages? These aspects are not discussed in the text.
  2. The description of the deep learning gesture recognition part is limited, and there is not much discussion on the experimental section or performance comparison with other mainstream models.

RESPONSE: Thank you for your suggestion. We have added the suggested content to the manuscript in section 2.5 (lines 254-270) of page 9 to establish a clearer focus.

  1. The conclusion lacks logical rigor and requires further modification.

RESPONSE: Thank you for pointing this out. We agree with you and have revised this suggestion throughout our conclusion.

Based on the reasons mentioned above, I will have to reject this paper.

CONCLUDING REMARKS: Again, thank you for giving us the opportunity to strengthen our manuscript with your valuable comments and queries. We have worked hard to incorporate your feedback and hope that these revisions persuade you to accept our submission.

Sincerely,

Yu-Ping Liao

Name: Yu-Ping Liao

Institute: Chung Yuan Christian University, Taiwan

Address: No. 200, Zhongbei Rd., Zhongli Dist., Taoyuan, 320

E-mail: lyp@cycu.org.tw

Round 2

Reviewer 1 Report

Comments and Suggestions for Authors

Some of the comments are addressed correctly. However, some other comments still need to be corrected. Please check the following comments:

1-Even though the sentence based on reference [1] is general, the source [1] still needs to be replaced with another reliable one.

2-Reference [2] is unpublished? Please replace it with another reliable reference. Same not for reference 18.

3- There are still some grammatical and linguistic errors that need to be corrected as mentioned previously. For example, the following issues:

-In line 49, it is stated that “and fast motion [12], and the paper also”, which paper? please be more clarify.

- There must be a linking sentence between the sentences in lines 23 and 33.

-The caption of figure 5 should be written on the same page with its related figure. In general, the entire presentation of the manuscript should be checked with the guidelines of the journal.

- It is preferable to avoid referring to the whole authors when mentioned them in the text body such as that in line 54 “D. Salami et al. [13] proposed using…”.

- please check the abbreviation rules. For example, in sub section 2.5, the same terms with their abbreviations are repeated in conclusion section. Please check such issues.

4- The two datasets mentioned in line 127 should be available for evaluation purposes. It was stated that all datasets are available. Are the links of database available?

5- Future works need to be more clarified.

6- The format of the references should be unified correctly, especially the new references add in the updated version.

Comments on the Quality of English Language

 Minor editing of English language requiredز

Author Response

Manuscript: Sensors ID -2772780
Response to Reviewer 1

1. Even though the sentence based on reference [1] is general, the source [1] still needs to be replaced with another reliable one.
RESPONSE: Thank you for pointing this out. The reviewer is correct, and we have replaced the literature [1] in section 1 of the manuscript.
2. Reference [2] is unpublished? Please replace it with another reliable reference. Same not for reference 18t.
RESPONSE: Thank you for pointing this out. The reviewer is correct, and we have replaced the literature [2] and [18] in section 1 of the manuscript.
3. There are still some grammatical and linguistic errors that need to be corrected as mentioned previously. For example, the following issues:
-In line 49, it is stated that “and fast motion [12], and the paper also”, which paper? please be more clarify.
- There must be a linking sentence between the sentences in lines 23 and 33.
-The caption of figure 5 should be written on the same page with its related figure. In general, the entire presentation of the manuscript should be checked with the guidelines of the journal.
- It is preferable to avoid referring to the whole authors when mentioned them in the text body such as that in line 54 “D. Salami et al. [13] proposed using…”.
- please check the abbreviation rules. For example, in sub section 2.5, the same terms with their abbreviations are repeated in conclusion section. Please check such issues..
RESPONSE: Thank you for providing these insights. We agree with you and have revised the sentences and terms incorporated in this suggestion throughout our paper.

4. The two datasets mentioned in line 127 should be available for evaluation purposes. It was stated that all datasets are available. Are the links of database available?
RESPONSE: Thank you for pointing this out. These two datasets are sample data that we have collected and are not currently available for open access.
5. Future works need to be more clarified.
RESPONSE: Thank you for providing these insights. We have added the suggested content to the manuscript in section 4 (lines 448-451) of page 18 to establish a clearer focus.
6. The format of the references should be unified correctly, especially the new references add in the updated version.
RESPONSE: Thank you for providing these insights. You have raised an important question. We have revised the sentences and terms incorporated in this suggestion throughout our paper.

CONCLUDING REMARKS: Again, thank you for giving us the opportunity to strengthen our manuscript with your valuable comments and queries. We have worked hard to incorporate your feedback and hope that these revisions persuade you to accept our submission.
Sincerely,
Yu-Ping Liao

Reviewer 2 Report

Comments and Suggestions for Authors

The resubmitted manuscript shows some merits for the  readers and researchers in hand gesture recognition, so my decision supports the publication.

Author Response

Manuscript: Sensors ID -2772780

Response to Reviewers

Dear Dr./Mr./Ms.: 

Thank you for inviting us to submit a revised draft of our manuscript entitled, " Implementation of a gesture recognition system with millimeter wave and thermal imager" to Sensors. We also appreciate the time and effort you and each of the reviewers have dedicated to providing insightful feedback on ways to strengthen our paper. Thus, it is with great pleasure that we resubmit our article for further consideration. We have incorporated most of the suggestions of reviewers. Please see below, in blue, for a point-by-point response to the reviewers’ comments and questions.

REVIEWER 2 COMMENTS:

The resubmitted manuscript shows some merits for the readers and researchers in hand gesture recognition, so my decision supports the publication.

RESPONSE: Thank you!

CONCLUDING REMARKS: Again, thank you for giving us the opportunity to strengthen our manuscript with your valuable comments and queries. Thank you and all the reviewers for the kind advice.

Sincerely,

Yu-Ping Liao

Reviewer 4 Report

Comments and Suggestions for Authors

The manuscript proposed a gesture recognition system with millimeter wave and thermal imager, the overall organization is well. The experimental result with 84% accuracy is ok but needs improvement in the real application.  Several issues:

1.  motivation is not clear.  COVID-19 is a special case, could it be used in other scenarios?   

2.  the algorithm speed is not illustrated?  The proposed method is favorable for real-time applications?

3. suggest adding some positive and negative recognition images, and giving more discussions 

4. in lines 420-421,  'Gated Recurrent Unit (GRU) outperforms the Long Short- 420 Term Memory (LSTM) and Recurrent Neural Network (RNN)'  just use abbreviations as you already defined in the previous.

5. some LSTM GRU-based approaches for animal/human behavior classification also can be referenced.  video-length may also influence recognition performance. 

Comments on the Quality of English Language

English  writing is ok 

Author Response

Manuscript: Sensors ID -2772780

Response to Reviewer 4

  1. motivation is not clear. COVID-19 is a special case, could it be used in other scenarios?  

RESPONSE: Thank you for pointing this out. The reviewer is correct, and we have added more recent and relevant literature to section 1 (lines 26-39) of the manuscript.

  1. the algorithm speed is not illustrated? The proposed method is favorable for real-time applications?

RESPONSE: Thank you for pointing this out. We have added the suggested content to the manuscript in section 3.3 of page 16 (lines 417-413) where the change can be found in the revised manuscript.

  1. suggest adding some positive and negative recognition images, and giving more discussions.

RESPONSE: We think this is an excellent suggestion. Thank you for this suggestion. We have added an extra section to discuss on page 18.

  1. in lines 420-421, 'Gated Recurrent Unit (GRU) outperforms the Long Short- 420 Term Memory (LSTM) and Recurrent Neural Network (RNN)'  just use abbreviations as you already defined in the previous..

RESPONSE: Thank you for your suggestion. We have revised the suggested content to the manuscript in section 5 of page 18 (lines 463-464) where the change can be found in the revised manuscript.

  1. some LSTM GRU-based approaches for animal/human behavior classification also can be referenced. video-length may also influence recognition performance.

RESPONSE: Thank you for pointing this out. In our future research work, we will refer to this literature to improve our study.

CONCLUDING REMARKS: Again, thank you for giving us the opportunity to strengthen our manuscript with your valuable comments and queries. We have worked hard to incorporate your feedback and hope that these revisions persuade you to accept our submission.

Sincerely,

Yu-Ping Liao
